# CROSS-MODAL FEW-SHOT LEARNING: A GENERATIVE TRANSFER LEARNING FRAMEWORK

## ABSTRACT

Most existing studies on few-shot learning focus on unimodal settings, where models are trained to generalize on unseen data using only a small number of labeled examples from the same modality. However, real-world data are inherently multi-modal, and unimodal approaches limit the practical applications of few-shot learning. To address this gap, this paper introduces the Cross-modal Few-Shot Learning (CFSL) task, which aims to recognize instances from multiple modalities when only a few labeled examples are available. This task presents additional challenges compared to classical few-shot learning due to the distinct visual characteristics and structural properties unique to each modality.

To tackle these challenges, we propose a Generative Transfer Learning (GTL) framework consisting of two stages: the first stage involves training on abundant unimodal data, and the second stage focuses on transfer learning to adapt to novel data. Our GTL framework jointly estimates the latent shared concept across modalities and in-modality disturbance in both stages, while freezing the generative module during the transfer phase to maintain the stability of the learned representations and prevent overfitting to the limited multi-modal samples. Our finds demonstrate that GTL has superior performance compared to state-of-the-art methods across four distinct multi-modal datasets: SKETCHY, TU-BERLIN, MASK1K, and SKSF-A. Additionally, the results suggest that the model can estimate latent concepts from vast unimodal data and generalize these concepts to unseen modalities using only a limited number of available samples, much like human cognitive processes.

## 1 INTRODUCTION

Collecting large amounts of labeled data in real-world applications is often prohibitively expensive, time-consuming, or simply impractical (Tharwat & Schenck, 2023; Sheng et al., 2024). Few-shot learning (FSL) has emerged as a viable solution, enabling models to generalize effectively using only a handful of labeled examples (Song et al., 2023; Chen et al., 2019; Luo et al., 2023; Ke et al., 2024). However, existing few-shot methods face significant challenges in handling the increasing prevalence of multi-modal data, such as multi-spectral images or multimedia content, which differs significantly from the extensive RGB data typically used in research. Multi-modal data, collected from multiple different types of sources and modalities such as different sensors or imaging protocols, is becoming increasingly essential in applications like surveillance (Wu et al., 2024; Hu et al., 2022) and medical image analysis (Jiang et al., 2023b; Mok et al., 2024). These challenges highlight the need for more advanced FSL frameworks capable of leveraging the complementary information inherent in multi-modal data (Luo et al., 2023; Jiang et al., 2023a).

Recent efforts have leveraged large pre-trained foundation models to extend their capabilities to novel multi-modal tasks, including tabular data (Ye et al., 2024), audio (Lin et al., 2023b; Duan et al., 2024), and video classification (Qing et al., 2023), moving beyond traditional unimodal tasks like image classification (Conti et al., 2023). Despite these advancements, the transfer of visual knowledge across different visual modalities remains relatively underexplored. Visual data, such as images and videos, constitute the most commonly studied data types; however, other visual modalities, such as infrared, depth, and sketches, exhibit both shared characteristics and distinct differences. These modalities share structural and contextual similarities with RGB data but possess unique attributes that make data collection and model adaptation more challenging.

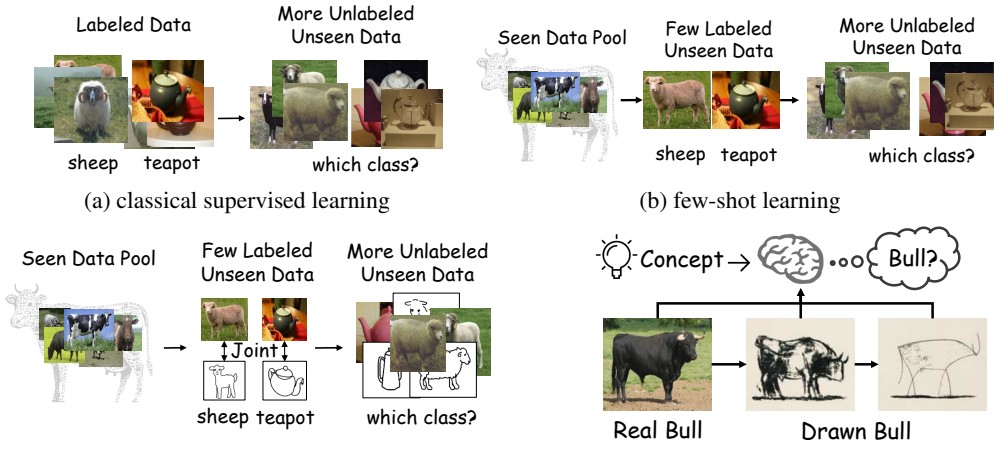

(a) classical supervised learning       (b) few-shot learning

(c) cross-modal few-shot learning (ours)       (d) generalizable visual concept

Figure 1: Comparison of recognition tasks. (a) Classical recognition requires extensive labeled data within a single modality. (b) Few-shot recognition uses a few labeled samples in a single modality to classify unseen samples. (c) Our proposed CFSL involves few labeled multi-modal samples and aims to generalize well to recognizing unseen multi-modal samples from the same classes, leveraging both seen and unseen data from different modalities. (d) Visual concept learning illustrates the ability to generalize concepts like "Bull" across real images and sketches (Pablo Picasso. *The Bull*, 1945.).

To address these challenges, this paper introduces a new Cross-modal Few-Shot Learning (CFSL) task, which aims to classify instances with multiple visual modalities using only a few labeled samples per class. In this task, the multi-modal data are organized into a support set and a query set. The support set contains a small number of labeled multi-modal examples per class, serving as the basis for learning. The query set includes unlabeled instances from the same classes and modalities that need to be classified. The ultimate goal is to train a model solely on the support set that can accurately classify the query set, regardless of modality. As illustrated in Figures 1(a-c), unlike classical supervised learning and FSL confined to a single modality, CFSL incorporates data from multiple visual modalities. The primary challenge of CFSL arises from the inherent variability and domain gaps between different visual modalities, which complicates feature extraction and alignment. Additionally, the task requires models to identify and leverage shared underlying semantics across modalities while adapting to the unique characteristics of each.

Previous studies have shown that compact visual representations are closely linked to real-world images, emphasizing the importance of underlying concepts in enabling humans to identify target objects (Vinker et al., 2022; 2023; Mukherjee et al., 2024). As illustrated in Figure 1d, these fundamental concepts can be readily learned with only a few examples (Lake & Piantadosi, 2020), thanks to the humans' ability to generalize from limited data, regardless of visual modality. This capability, referred to as "evocation", is grounded in the vast reservoir of previously encountered visual experiences that support recognition and generalization. Inspired by this observation, we hypothesize that models can achieve similar evocative capabilities by learning latent concepts from abundant unimodal data.

We propose a Generative Transfer Learning (GTL) framework to facilitate knowledge transfer between unimodal and multi-modal data. In the GTL framework, we posit that the latent concepts underlying target objects consist of two components: (1) an intrinsic concept that captures the core characteristic shared across modalities, and (2) an in-modality disturbance that accounts for variations unique to each modality. Our approach aims to estimate these latent components and encode the relationships between the intrinsic concepts and visual content. By disentangling the intrinsic concepts from the in-modality disturbances, the GTL framework enables adapting to multi-modal data while preserving the transferable relationships learned from unimodal data, thereby improving both adaptability and accuracy across various modalities.

Within the GTL framework, our methodology consists of a two-stage training process. In the first stage, the **generative learning stage**, the model learns latent concepts label-free, relying solely on pre-trained visual representations. This stage focuses on capturing the intrinsic concept and the variations in visual content across modalities, ensuring that the learned relationship between the latent

concept and visual content is robust and transferable. In the second stage, the **recognition stage**, the backbone network is frozen, and a separate classifier is trained on top of the learned latent intrinsic concepts to perform label prediction. This two-stage framework enables the model to disentangle intrinsic visual concepts from in-modality disturbances, facilitating effective recognition across visual modalities.

Our contributions are summarised as follows: (i) We introduce a new cross-modal few-shot learning task, which focuses on the connection and distinction between visual modalities, requiring models to perform recognition on multi-modal data with minimal labeled samples. This task better reflects real-world scenarios, where multi-modal data is scarce and diverse. (ii) We propose the generative transfer learning framework, designed to disentangle intrinsic concepts from in-modality disturbances, enabling efficient knowledge adaption across modalities. (iii) We demonstrate the effectiveness of our method through extensive experiments on multiple cross-modal datasets, including SKETCHY (Sangkloy et al., 2016), TU-BERLIN (Eitz et al., 2012), the fine-grained biometric dataset MASK1K (Lin et al., 2023a) and SKSF-A (Yun et al., 2024).

## 2 TASK SETTINGS

In this section, we formally define the proposed CFSL task and highlight its differences from the previous FSL task. Additionally, we provide an overview of the CFSL task, emphasizing the challenges posed by handling multiple visual modalities with limited labeled data.

**Dataset Setup**  For the proposed CFSL task, the dataset $D = \{(x_m^i, y^i), y^i \in Y\}$ comprises a **base unimodal dataset** $D_{\text{base}} = \{(x_m^i, y^i), y^i \in Y_{\text{base}}, m = 1\}$ and a **novel multimodal dataset** $D_{\text{novel}} = \{(x_m^i, y^i), m \in \{m^1, m^2, \ldots, m^d\}, y^i \in Y_{\text{novel}}\}$. Here, $x_m^i$ denotes the feature vector of the $i$-th sample in modality $m$, $y^i$ is the corresponding class label. $Y_{\text{base}}$ and $Y_{\text{novel}}$ represent the set of class labels for the base and novel datasets, respectively. Notably, the class labels between the base and novel datasets are disjoint, aka, $Y_{\text{base}} \cap Y_{\text{novel}} = \varnothing$, such that $Y_{\text{base}} \cup Y_{\text{novel}} = Y$.

In classical FSL tasks, the goal is to train a model on $D_{\text{base}}$ with labels $Y_{\text{base}}$ and transfer this knowledge to improve the recognition performance on novel classes $Y_{\text{novel}}$ with the same modality, using only a few labeled examples for each class. In contrast, our CFSL task introduces the added challenge of handling multi-modal data in the novel dataset $D_{\text{novel}}$ while the base dataset $D_{\text{base}}$ contains only unimodal data (e.g., RGB images). As shown in Figure 2a, there is a clear boundary among the multi-modal data. This setting reflects real-world scenarios where models are typically trained on unimodal data but are expected to generalize effectively to multi-modal data.

The novel dataset $D_{\text{novel}}$ is further divided into:

- A support set $D_{\text{support}} = \{(x_m^i, y^i)|m \in \{m^1, m^2, \ldots, m^d\}, y^i \in Y_{\text{novel}}\}$, containing a limited number of labeled samples per class from multiple modalities. Typically, this consists of $K$ labeled samples for each of $N$ classes.

- A query set $D_{\text{query}} = \{x_m^j|m \in \{m^1, m^2, \ldots, m^d\}\}$, containing a larger number of unlabeled samples from the same classes as $D_{\text{support}}$, but potentially from the different modalities.

The fundamental challenge in this task is the limited labeled data in $D_{\text{support}}$, combined with the multi-modal nature of the data in $D_{\text{novel}}$. The scarcity of labeled data per class, along with the need to generalize across different modalities, makes this task significantly more challenging.

**Task Overview**  To classify the samples in $D_{\text{query}}$, the proposed CFSL involves a feature extraction function $e_\Phi$, which is initially pre-trained on the large and fully labeled base dataset $D_{\text{base}}$. The goal is to adapt $e_\Phi$ to $e_{\Phi'}$, allowing it to extract discriminative features from all modalities in $D_{\text{novel}}$ for classification. The adapted feature extractor $e_{\Phi'}$ should ensure that samples from the same class $y^i$ are mapped close to each other in the feature space, enabling accurate classification regardless of the modality $m$. This adaptation process should help the classifier $c_\phi$, parameterized by $\phi$, in predicting the correct class labels for samples in $D_{\text{query}}$ using features produced by $e_{\Phi'}$. The adaptation allows the model to generalize across the novel modalities and classes, even with very few labeled examples in the support set.

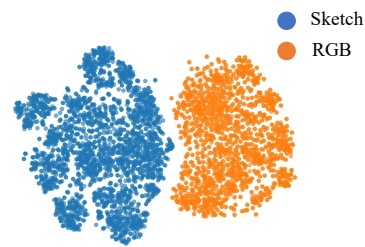
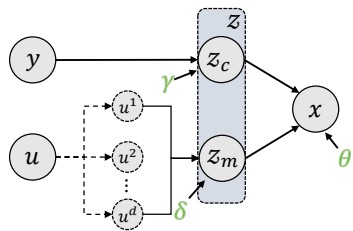

(a) t-SNE of feature distribution of different modalities. (b) The data generating model for CFSL.

Figure 2: The observation of the severe modality differences and the details of the proposed generative model. (a) Illustration of the modality difference by the t-SNE clustering of the pre-trained CLIP (Radford et al., 2021) features of different modalities from Mask1K dataset (Lin et al., 2023a). (b) The proposed generative process for the representation learning stage, the green symbols are assumed to be parameters that enable the models to adapt from base to novel data.

## 3 METHODOLOGY

This section formulates the problem, discusses network design, and introduces the learning objectives.

### 3.1 PROBLEM FORMULATION

We assume that each observation $\mathbf{x}$ is generated from a nonlinear function $\mathbf{g}$:

$$\mathbf{x} = \mathbf{g}(\mathbf{z}) = \mathbf{g}(\mathbf{z}_m, \mathbf{z}_c),$$

where $\mathbf{z} = (\mathbf{z}_m, \mathbf{z}_c)$, $\mathbf{z}_m$ contains in-modality disturbance, and $\mathbf{z}_c$ encapsulates latent intrinsic concept.

Figure 2b illustrates our data generating process. We formalize the probabilistic joint distribution of our data generating process by:

$$p(\mathbf{x}, \mathbf{z}, \mathbf{u}, y) = p_\delta(\mathbf{z}_m | \mathbf{u}^1, \mathbf{u}^2, \ldots, \mathbf{u}^d) p_\gamma(\mathbf{z}_c | y) p_\theta(\mathbf{x} | \mathbf{z}_m, \mathbf{z}_c) p(\mathbf{u}) p(y). \tag{1}$$

We use a VAE to model the generator $p_\theta(\mathbf{x}|\mathbf{z}_m, \mathbf{z}_c)$, where $\theta$ are the parameters, and $(\mathbf{z}_m, \mathbf{z}_c)$ are obtained by encoding $\mathbf{x}$ with parameters $\alpha$ via the posterior estimator $q_\alpha(\mathbf{z}|\mathbf{x})$. The $\delta$ and $\gamma$ are the parameters for modeling the distributions of $\mathbf{z}_m$ (via $p_\delta(\mathbf{z}_m|\mathbf{u}^1, \mathbf{u}^2, \ldots, \mathbf{u}^d)$) and $\mathbf{z}_c$ (via $p_\gamma(\mathbf{z}_c|y)$), respectively. To learn domain variable $\mathbf{u}$ and predict the correct label $y$, we also introduce two additional modules: a disturbance encoder $q_\eta(\mathbf{u}|\mathbf{x})$ with parameters $\eta$, and a classifier $q_\phi(\hat{y}|\mathbf{z}_c)$ with parameters $\phi$. In what following, we will introduce each component in detail.

First, we introduce a posterior estimator parameterized by $\alpha$ as the encoder to learn the visual latent representation $\mathbf{z} = \{\mathbf{z_c}, \mathbf{z_m}\}$ as:

$$\mathbf{z} \sim q_\alpha(\mathbf{z}|\mathbf{x}). \tag{2}$$

Second, since we lack supervision for learning the in-modality disturbance, we assume that this information can be estimated from the observed data $\mathbf{x}$. Therefore, a disturbance estimator, parameterized by $\eta$, is introduced to estimate the modality-relevant latent variable $\mathbf{u}$ as:

$$\mathbf{u} \sim q_\eta(\mathbf{u}|\mathbf{x}), \tag{3}$$

where the multi-perspective domain variables $\mathbf{u}^1, \mathbf{u}^2, \ldots, \mathbf{u}^d$ are generated from $\mathbf{u}$, capturing different perspectives of the modality.

Third, the modality-specific latent variable $\mathbf{z}_m$ is derived from the multi-perspective latent variables $\mathbf{u}^1, \mathbf{u}^2, \ldots, \mathbf{u}^d$, governed by the learnable parameter $\delta$:

$$\mathbf{z}_m \sim p_\delta(\mathbf{z}_m | \mathbf{u}^1, \mathbf{u}^2, \ldots, \mathbf{u}^d) \tag{4}$$

Fourth, the latent intrinsic concept variable $\mathbf{z}_c$, which captures class-specific concept information, depends only on class label $y$ and is governed by the parameter $\gamma$:

$$\mathbf{z}_c \sim p_\gamma(\mathbf{z}_c | y) \tag{5}$$

Figure 3: The proposed GTL framework. During the training on base data, all modules are trained (as in the blue dashed box), but when adapting to novel data, the generator is frozen, and all other parts are tunable (as in the red dashed box). The classifier for recognition is separately initialized on base and novel training since there are non-overlap classes between them.

Finally, the generator conceptualizes each observation $\mathbf{x}$ as being derived from a non-linear smooth mixing transformation, parameterized by $\theta$, involving latent variables $\mathbf{z} \subseteq \mathcal{Z} \in \mathbb{R}^n$, which are decomposed into two components $\mathbf{z}_c \in \mathbb{R}^{n_c}$ and $\mathbf{z}_m \in \mathbb{R}^{n_m}$, and

$$\mathbf{x} \sim p_\theta(\mathbf{x}|\mathbf{z}_m, \mathbf{z}_c) \tag{6}$$

In the context of CFSL, the differing distributions of modality-relevant information $\mathbf{u}$ and class label $y$ between base and novel datasets necessitate adapting model parameters, including the encoder $q_\alpha(\mathbf{z}|\mathbf{x})$, the disturbance estimator $q_\eta(\mathbf{u}|\mathbf{x})$, and the parameters responsible for generating the latent variables $\mathbf{z}_m$ and $\mathbf{z}_c$. However, the non-linear transformation, parameterized by $\theta$, remains invariant during the transfer learning stage, as it is assumed to capture the stable relationship between the latent concept and visual content. This invariance ensures that the model can adapt to novel data while preserving key generalizable components.

## 3.2 NETWORK DESIGN

In this section, we introduce the key components of our network, each playing a distinct role in modeling the cross-modal few-shot classification task.

**Encoder**  We denote the encoder that performs posterior estimation as $q_\alpha(\hat{\mathbf{z}}|\mathbf{x})$, where $\hat{\mathbf{z}}$ represents the estimated latent variables. We assume that the two components $\mathbf{z}_m$ (modality-specific) and $\mathbf{z}_c$ (class-relevant) are conditionally independent given the observation $\mathbf{x}$, allowing us to factorize the posterior distribution as:

$$q_\alpha(\mathbf{z}|\mathbf{x}) = q_\alpha(\mathbf{z}_m|\mathbf{x})q_\alpha(\mathbf{z}_c|\mathbf{x}) \tag{7}$$

Accordingly, we approximate the joint posterior distribution by assuming an isotropic Gaussian, characterized by a mean $\mu$ and covariance $\sigma^2$, as follows:

$$q_\alpha(\mathbf{z}_c, \mathbf{z}_m|\mathbf{x}) \sim \mathcal{N}(\mu, \sigma^2) \tag{8}$$

To learn this posterior distribution, we employ 1-layer MLPs with ReLU activation, a batch normalization layer, and a dropout layer as the estimator.

**Disturbance Encoder**  The in-modality disturbance latent variable $\mathbf{z_m}$ is assumed to be transferable when adapting from the base to novel data. Since there is a lack of direct supervision regarding which representation corresponds to modality-specific information, we employ a flexible estimator to approximate the unobservable prior $p(\mathbf{z}_m)$. Inspired by the latent domain learning method (Deecke et al., 2022), we use a set of learnable gating functions $g(\mathbf{x})$ that assign each observation $\mathbf{x}$ to multiple latent domains, capturing different perspectives.

The estimated modality-relevant variable $\hat{\mathbf{u}}$ is used to guide the modality-specific variable $\hat{\mathbf{z}}_m$, and the estimation is given by:

$$\hat{\mathbf{z}}'_m = h_{\delta, \hat{\mathbf{u}} \sim q_\eta}(\hat{\mathbf{u}}, \hat{\mathbf{z}}_m) \quad \text{and} \quad q_\eta(\hat{\mathbf{u}}|\mathbf{x}) = \sum_{d=1}^{D} g_d(\mathbf{x})V_d(\mathbf{x}), \tag{9}$$

where $d$ is a hyperparameter that determines the number of latent domains, and each $V_d(\mathbf{x})$ is parameterized through a linear transformation. The function $h(\mathbf{u}, \mathbf{z_m})$ represents a linear aggregation of the domain variable $\mathbf{u}$ and the latent variables $\mathbf{z_m}$, with trainable parameter $\delta$.

**Reconstruction** The reconstruction module is responsible for generating an estimate of the observation $\hat{\mathbf{x}}$ based on the estimated latent variables $\hat{\mathbf{z}}_\mathbf{c}$ (class-relevant information) and $\hat{\mathbf{z}}'_\mathbf{m}$ (modality-relevant information). We adopt a generator with a structure similar to, but reverse of, the posterior estimator. While the posterior estimator encodes the latent variables, the generator decodes them back into the observed data.

The conditional distribution $p_\theta(\mathbf{x}|\mathbf{z}_c, \mathbf{z}'_m)$ is modeled by the generator, which consists of a 1-layer MLP with a ReLU activation function, a batch normalization layer, and a dropout layer. The reconstruction process is formulated as:

$$\hat{\mathbf{x}} = \text{Dropout}\left(\text{ReLU}\left(\text{MLP}(\hat{\mathbf{z}}')\right)\right) \quad \text{where} \quad \hat{\mathbf{z}}' = \{\hat{\mathbf{z}}_c, \hat{\mathbf{z}}'_m\} \tag{10}$$

**Classification** The classifier $p_\phi(\mathbf{y}|\mathbf{z}_c)$, parameterized by $\phi$, can be implemented using a simple linear classifier. This involves a weight matrix $\mathbf{W} \in \mathbb{R}^{d \times c}$, where $d$ is the dimension of the latent variable $\hat{\mathbf{z}}_c$, and $c$ denotes the number of output classes. The classifier is trained using a standard cross-entropy loss function. The linear classifier computes the logits directly by linearly combining the latent variable $\hat{\mathbf{z}}_c$ with the weight matrix $\mathbf{W}$, followed by a softmax function to output class probabilities. The logits are computed as:

$$\hat{\mathbf{y}} = \text{softmax}(\mathbf{W}^\top \hat{\mathbf{z}}_c), \tag{11}$$

where $\hat{\mathbf{y}}$ represents the predicted class probabilities.

For detailed network architecture, please refer to Appendix C.

### 3.3 Learning Objectives

**Representation learning** Based on the above generative learning structure, the training objectives in the representation learning phase are formulated by the evidence lower bound (ELBO) as follows:

$$\mathcal{L}_{\text{ELBO}} = \underbrace{\mathbb{E}_{q_{\hat{\mathbf{z}}_c, \hat{\mathbf{z}}'_m | \mathbf{x}}}\left[\ln p_\theta(\mathbf{x}|\hat{\mathbf{z}}_c, \hat{\mathbf{z}}'_m)\right]}_{\text{Reconstruction Loss}} - \lambda \underbrace{\mathbb{E}_{\hat{\mathbf{z}}_c, \hat{\mathbf{z}}'_m \sim q_\alpha, q_\eta, h_\delta}\left[\log q(\hat{\mathbf{z}}_c, \hat{\mathbf{z}}'_m | \mathbf{x}) - \log p(\mathbf{z})\right]}_{\text{KL Divergence}}, \tag{12}$$

where the reconstruction term ensures that the model accurately reconstructs the input data from the latent variables $\mathbf{z}_c$ and $\mathbf{z}_m$, while the KL divergence term regularizes the latent space to ensure the learned features align with the underlying data distribution. The hyperparameter $\lambda$ controls the trade-off between reconstruction accuracy and regularization strength.

**Classification learning** In this phase, the cross-entropy loss measures the discrepancy between the predicted and actual class labels.

$$\mathcal{L}_{\text{CE}} = -\mathbb{E}_{\hat{\mathbf{y}}}(\mathbf{y} \log \hat{\mathbf{y}}). \tag{13}$$

### 3.4 Training and Inference

The operational framework of GTL is depicted in Figure 3, details the training and adaptation process for CFSL scenarios. This section outlines the workflow, from initial training on the base dataset to subsequent adaptation for novel data, clarifying the methodologies employed in each phase.

**Phase 1: Training on Base Data** The initial training phase is fundamental as it establishes the distinction between transferable and non-transferable components within the model. During this phase, all modules are trained using the base dataset $D_{\text{base}}$. Specifically, the posterior estimator ($\alpha$), the disturbance estimator and aggregator ($\eta$ and $\delta$), and the generator ($\theta$) are jointly trained using Eq. 12. Afterward, the classifier ($\phi$) is trained using Eq. 13. The primary goal is to robustly encode domain-specific variations and content-specific features into separate latent spaces.

**Phase 2: Transfer Learning for Adapting to Novel Data**   In this phase, the model is exposed to a novel data support set $D_{\text{support}}$. Given our assumption that the relationship between latent representations and visual content remains consistent across both base and novel datasets, we freeze the generator ($\theta$) in its trained state. This decision ensures that the foundational decoding process, which reconstructs visual content from latent representations, remains stable and unaffected by new data variability. We first fine-tune the posterior estimator ($\alpha$), the disturbance estimator, and aggregator ($\eta$ and $\delta$) using Eq. 12, and then update the classifier ($\phi$) with a few labeled examples using Eq. 13.

**Inference with Novel Data Query Set**   In the inference phase, samples from the novel data query set $D_{\text{query}}$ are first processed by our posterior estimator ($\alpha$), which generates latent intrinsic concepts based on the learned model. These representations are then used by the trained classifier ($\phi$) to make predictions. We select the prediction $\hat{y}$ with the maximum value, which corresponds to the highest predicted probability across all possible classes, to determine the most likely class label.

## 4 EXPERIMENTS

The experiments section consists of Experimental Setup and Benchmark Results. The setup covers datasets, split strategy, evaluation protocol, and implementation details. The results include a summary of outcomes, an ablation study, and a discussion to verify our hypotheses.

### 4.1 EXPERIMENTAL SETUP

**Datasets**   We conduct experiments on multiple cross-modal datasets, including two object multi-modal datasets and two biometric multi-modal datasets. Given that sketch is the most common and accessible visual modality beyond RGB, we use multi-modal datasets consisting of both RGB and sketch data to verify the proposed method. For the object multi-modal datasets, SKETCHY (Sangkloy et al., 2016) contains 75,471 sketches and 73,002 images across 125 categories, while TU-BERLIN (Eitz et al., 2012; Zhang et al., 2016) includes 20,000 sketches and 204,489 images spanning 250 categories. For the more challenging biometric multi-modal datasets which exhibit less inter-class variance, MARKET-SKETCH-1K (MASK1K) (Lin et al., 2023a) contains over 4,700 sketches representing 996 identities in 6 styles, paired with 20,480 matching photographs from MARKET1501 (Zheng et al., 2015). Additionally, we use SKSF-A (Yun et al., 2024), consists of 938 face-sketch pairs of 134 identities across 7 distinct styles.

**Dataset Split**   For experiments on SKETCHY, we follow the dataset splitting scheme used by Bhunia et al. (2022), which contains 64 base classes, while the remaining 61 classes are used as the novel set to make the task more challenging. For TU-BERLIN, we split into 125 base and 125 novel classes following Bandyopadhyay et al. (2024). For the smaller biometric multi-modal dataset, all data in MASK1K and SKSF-A are considered novel; we use commonly adopted person and face datasets MSMT17 (Wei et al., 2018) and CelebA-HQ (Karras et al., 2018) as base sets for person and face recognition, respectively. Each base set is further divided into training, validation, and testing subsets (60%:20%:20%) following Bhunia et al. (2022).

**Evaluation Protocol**   We report the results under two settings: **all-way-$k$-shot** (Ju et al., 2022; Li et al., 2024b) and **standard 5-way-$k$-shot** (Luo et al., 2023). In the all-way-$k$-shot setting, all classes are presented to the model, with $k$ examples randomly sampled per class to form the support set $S$, while the remaining examples constitute the query set $Q$. In the 5-way-$k$-shot setting, 5 classes are randomly selected per episode, and for each of these 5 classes, $k$ examples are sampled to form the support set $S$, with the remaining examples used for the query set $Q$.

In both settings, $k$ denotes the number of labeled samples per class in the support set $S$. After splitting the data into a base set and a novel set comprising $S \cup Q$, $k$ determines how many samples per class are included in $S$, with the remaining samples forming $Q$.

**Evaluation metric**   We compare model performance using **Top-1 Accuracy** over the novel query set, referred to as $Acc@avg$, which measures the proportion of instances where the model's top predicted label matches the true label across mixed modalities. Additionally, we report the average Top-1 accuracy within each modality.

Table 1: Quantitative results of ours and other sota competitors on SKETCHY dataset. The best results are marked as **BOLD.** The A(B/C) metrics under the *Acc@avg* stands for the average Top-1 accuracy for the A: mixed-modality, B: sketch, and C: RGB data.

| Methods | all-way-1-shot | all-way-5-shot | 5-way-1-shot | 5-way-5-shot |
| | *Acc@avg* | *Acc@avg* | *Acc@avg* | *Acc@avg* |
| --- | --- | --- | --- | --- |
| ICML23 | 27.6 (29.9 / 25.1) | 33.8 (18.1 / 50.6) | 28.8 (24.9 / 33.1) | 33.7 (16.8 / 52.4) |
| C2-Net | 40.2 (24.7 / 56.7) | 42.0 (26.5 / 58.6) | 44.2 (40.8 / 48.0) | 57.8 (50.1 / 66.3) |
| AGW | 21.1 (23.9 / 18.2) | 46.0 (50.4 / 41.4) | 45.2 (47.0 / 43.2) | 71.7 (82.8 / 59.1) |
| TransReID | 46.0 (23.4 / 70.2) | 64.3 (59.9 / 67.0) | 78.6 (64.6 / 94.0) | 89.6 (83.4 / 96.4) |
| CLIP-ReID | 60.7 (56.2 / 65.4) | 81.4 (77.9 / 85.0) | 83.2 (76.9 / 90.1) | 94.0 (95.3 / 92.5) |
| Ours | **63.8 (58.8 / 69.1)** | **82.9 (79.8 / 86.3)** | **84.5 (81.3 / 87.9)** | **94.1 (95.4 / 92.9)** |

Table 2: Quantitative results of ours and other sota competitors on TU-BERLIN dataset.

| Methods | all-way-1-shot | all-way-5-shot | 5-way-1-shot | 5-way-5-shot |
| | *Acc@avg* | *Acc@avg* | *Acc@avg* | *Acc@avg* |
| --- | --- | --- | --- | --- |
| ICML23 | 22.9 (34.3 / 21.8) | 48.8 (31.5 / 50.4) | 35.5 (47.4 / 34.2) | 50.9 (48.0 / 51.2) |
| C2-Net | 39.1 (28.8 / 40.1) | 50.1 (46.9 / 50.4) | 60.1 (60.9 / 70.0) | 65.3 (74.9 / 74.3) |
| AGW | 14.7 (32.0 / 13.0) | 40.1 (64.2 / 37.9) | 69.8 (79.5 / 68.8) | 75.9 (89.6 / 74.5) |
| TransReID | 38.0 (29.9 / 38.8) | 61.1 (68.9 / 60.3) | 83.5 (63.3 / 85.7) | 93.3 (92.8 / 93.4) |
| CLIP-ReID | 46.6 (44.2 / 46.8) | 74.7 (75.8 / 74.6) | 91.5 (79.7 / 92.8) | 97.1 (94.9 / 97.3) |
| Ours | **47.1 (46.3 / 47.3)** | **74.8 (76.1 / 74.7)** | **92.2 (85.1 / 93.0)** | **98.0 (98.1 / 98.0)** |

**Implementation Details**  We implement our framework in PyTorch and run experiments on an NVIDIA RTX 2080Ti GPU. ViT-B/16 (Dosovitskiy et al., 2021), pre-trained on the base data with CLIP (Radford et al., 2021) for backbone initialization as the visual encoder to extract the visual representation $\mathbf{x}$. The hyperparameter $d$ is set to 128, and $\lambda$ is set to 1. We use the Adam optimizer Kingma & Ba (2015) for 60 epochs with an initial learning rate of $1e^{-3}$ during the representation learning stage and $1e^{-4}$ during the classification stage. The learning rate decreases to 10% of the original value after 30 epochs, and weight decay is fixed at $1e-4$ for all settings.

## 4.2  BENCHMARK RESULTS

We benchmark our method against several few-shot learning methods, including C2-Net (Ma et al., 2024) and ICML23 (Luo et al., 2023), as well as fine-grained retrieval models AGW (Ye et al., 2021), TransReID (He et al., 2021), and CLIP-ReID (Li et al., 2023b).

**Comparison on multi-modal category datasets**  *Sketchy*: Table 1 presents the quantitative results on the SKETCHY dataset. Our method consistently outperforms all five leading benchmarks, with the highest improvement reaching 4.4% in accuracy across different settings. Notably, in the all-way-1-shot setting, our approach achieves a 3.1% improvement in overall accuracy compared to the best-performing method. Additionally, our model shows gains in both individual modality classification metrics (sketch and RGB) across all settings.

*TU-Berlin*: The TU-BERLIN dataset, which incorporates more abstract representations of data and has a larger scale than SKETCHY, but contains fewer sketches. As shown in Table 2, despite the increased difficulty due to the larger number of classes and more challenging multi-modal samples, the proposed GTL method consistently achieves the best performance across all accuracy metrics. Notably, in the 5-way-$k$-shot setting, our approach delivers a 5.4% improvement in sketch modality accuracy, further demonstrating its robustness in handling cross-modal few-shot recognition tasks.

**Comparison on biometric multi-modal datasets**  Given the limited modality data per class (8 samples for both MASK1K and SKSF-A), we decrease the $k$ to 1 and 2 and evaluate the models only under the all-way settings.

*Mask1K*: Figures 4a and 4b present the experimental results on the MASK1K dataset. The increased number of classes and the scarcity of training samples pose significant challenges for existing methods.

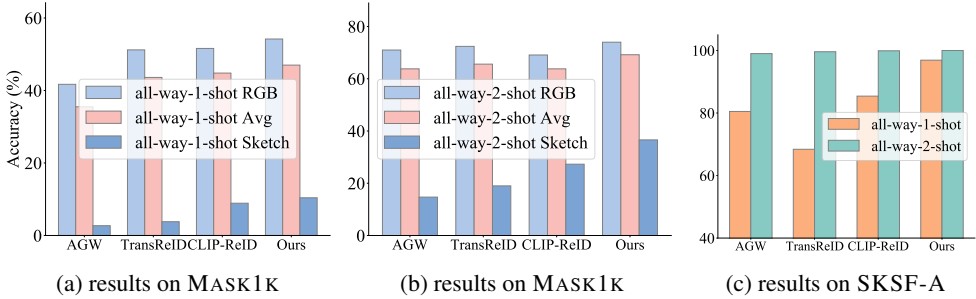

Figure 4: Experimental results of different shots on testing performance on (a) and (b) MASK1K, and (c) SKSF-A datasets.

As shown in the figures, despite the biometric differences being harder to capture, our proposed GTL method achieved notable accuracy improvements of 2.2% and 5.4% for $k = 1$ and $k = 2$, respectively. Significant improvements were also observed in individual modality classification metrics.

*SKSF-A:* Figure 4c illustrates the performance on the SKSF-A dataset. In this experiment, we report only the average accuracy for the sketch modality, as there is only one RGB image per class available for fine-tuning. The results demonstrate that, regardless of $k = 1$ or $k = 2$, all methods achieved relatively high accuracy, indicating clear inter-class separability in the dataset. Notably, our method showed an improvement of 11.5% for $k = 1$, highlighting its ability to adapt effectively to limited data while maintaining strong performance.

### 4.3 ABLATION STUDIES

To verify the effectiveness of each module in GTL framework, Table 3 presents the the ablation study results. The first two rows show the results of removing components: "w/o $\mathbf{z}$" excludes all latent variables, using only the classifier ($\phi$) for label prediction; "w/o $\mathbf{z_m}$" removes only the disturbance latent variable $\mathbf{z_m}$. The lower rows compare training strategies. GTL$_T$ trains all modules from scratch, while GTL$_{FT}$ fine-tunes the generator during adaptation instead of keeping it fixed. The last row represents our complete GTL framework, achieving the best results in both settings, as indicated by the bolded accuracy scores.

Table 3: Ablation studies of each component in GTL on SKETCHY dataset.

|  | all-way-1-shot | all-way-5-shot |
|---|---|---|
|  | *Acc@avg* | *Acc@avg* |
| w/o $\mathbf{z}$ | 48.7 (40.0 / 58.0) | 67.8 (59.2 / 76.9) |
| w/o $\mathbf{z}_m$ | 53.9 (41.7 / 67.0) | 73.8 (63.9 / 84.3) |
| GTL | **63.8 (58.8 / 69.1)** | **82.9 (79.8 / 86.3)** |
| GTL$_T$ | 44.2 (40.7 / 48.0) | 69.9 (65.9 / 74.0) |
| GTL$_{FT}$ | 61.5 (56.6 / 66.8) | 81.1 (78.1 / 84.4) |
| GTL | **63.8 (58.8 / 69.1)** | **82.9 (79.8 / 86.3)** |

Comparing the results in Table 3, incorporating latent concept learning significantly improves performance. Including $\mathbf{z_m}$ (third row) increases RGB accuracy by 18.8% and sketch accuracy by 11.1% over the baseline (first row). Excluding modality-specific variables ("w/o $\mathbf{z_m}$", second row) improves RGB accuracy by 9% and sketch accuracy by 1.7%. Fixing the generator after pre-training (final row) results in the highest average accuracy of 19.6%, outperforming the variant where the generator is fine-tuned. Additional information regarding selecting the hyperparameter of the learnable latent domain number $d$ is provided in Appendix B.

### 4.4 DISCUSSIONS

Beyond the ablation studies, this section examines our hypothesis from data distribution perspectives.

**Assumption validation: Concept transfer** To validate the invariance of the relationship between latent concepts and visual content and the estimability of in-modality disturbance, we analyzed data distributions. We trained models on base unimodal data, novel multimodal mixed data, RGB data, and sketch data. As shown in Figure 5, pair plots of the generator's outputs depict the relationship between latent variables $\hat{\mathbf{z}}$ and visual representations $\mathbf{x}$. Significant overlap between base data and novel data distributions supports our assumption. Notably, the overlap with sketch data is greater than with RGB data from the novel set, likely due to differences in underlying data distributions and content diversity. Sketches capture fundamental structures common to the base data, leading to closer alignment in the latent space.

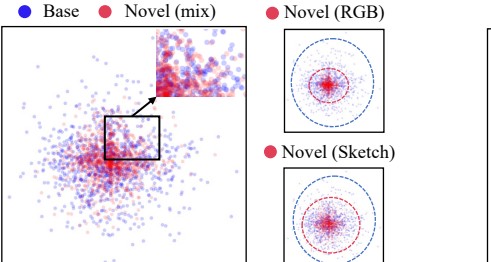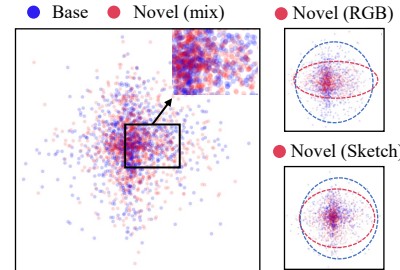

Figure 5: Visualizations of the pair plot of the learned latent representation and the visual representation on SKETCHY (left) and MASK1K (right) datasets. Blue stands for model trained on based data, and red denotes the model trained the novel data. The dashed circle with the corresponding color denotes the approximate scope of distributions. The (mix) denotes the trained data are multi-modal, and (RGB) and (Sketch) denote the unimodal data are used for training.

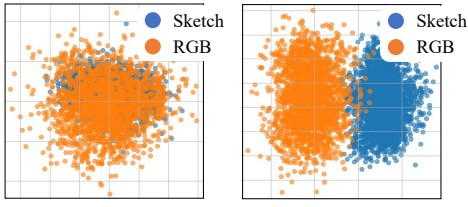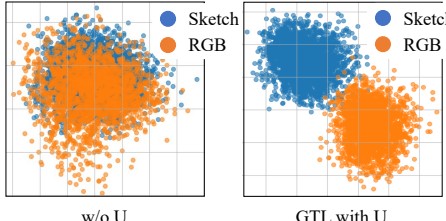

Figure 6: The t-SNE of the estimated in-modality disturbance representation on SKETCHY (left) and MASK1K (right) datasets.

**Assumption validation: Estimating in-modality disturbance**    To further assess our assumption about estimating in-modality disturbances, we performed t-SNE clustering of the latent representations $\hat{u}$ from models trained with and without the disturbance estimator. As shown in Figures 6, without the disturbance estimator, latent representations from different modalities—such as RGB images and sketches—are intermixed in the latent space, indicating the model struggles to differentiate modality-specific features. In contrast, when the disturbance estimator is included, t-SNE visualizations reveal distinct clusters for each modality. This demonstrates that the estimator effectively separates modality-specific disturbances and preserves unique characteristics within the latent space.

## 5    CONCLUSIONS

In this work, we tackled the limitations of unimodal few-shot learning by introducing the cross-modal few-shot learning task, which addresses real-world scenarios involving multiple visual modalities with only a few labeled examples. Unlike classical few-shot learning, our task presents additional challenges due to the inherent variability in visual characteristics, structural properties, and domain gaps between modalities.

To overcome these challenges, we proposed the generative transfer learning framework, designed to enable efficient knowledge transfer from abundant unimodal data to data-scarce multi-modal scenarios. By estimating shared latent concepts from unimodal data and generalizing them to unseen modalities, the GTL framework effectively disentangles modality-independent representations from in-modality disturbances. Our experimental results demonstrated that our GTL framework significantly outperforms state-of-the-art methods on four multi-modal datasets: Sketchy, TU-Berlin, Mask1K, and SKSF-A.

**Limitations:**    While our approach demonstrates strong performance, the lack of diverse visual multi-modal datasets for visual recognition remains a significant challenge. Most existing datasets focus on a limited number of visual modalities (e.g., RGB and sketch), restricting the evaluation of models designed to handle more complex visual data. Future work will require more extensive and varied multi-modal datasets to fully explore and validate the potential of cross-modal few-shot learning in a broader range of visual contexts.

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

# A RELATED WORK

## A.1 RELATION TO UNIMODAL FEW-SHOT LEARNING

In general, unimodal learning tasks can be broadly categorized into three types based on the availability and quantity of labeled data: (i) **Supervised Learning**, where a large amount of labeled data is available for training, allowing models to learn features and perform accurate recognition (Yan et al., 2023a; Han et al., 2023; Li et al., 2023a); (ii) **Few-shot Learning**, where only a limited number of labeled samples are provided for each class, challenging the model to generalize effectively from minimal data (Chen et al., 2019; Luo et al., 2023); and (iii) **Zero-shot Learning**, where no labeled examples are available for certain classes (Wei et al., 2023; Li et al., 2024a; Mirza et al., 2024).

Specifically, Few-shot learning (FSL) encompasses a training phase where a model is trained on a relatively large dataset and an adaptation phase in which the trained model is adjusted to previously unseen tasks with limited labeled samples. Most existing FSL tasks utilize unimodal datasets for training and testing, including popular benchmarks such as ImageNet (Deng et al., 2009), CIFAR (Oreshkin et al., 2018), CUB-200-2021 (Wah et al., 2011), and Stanford Dogs (Khosla et al., 2011). FSL typically involves three main approaches: (i) **Meta learning** (Sun et al., 2019; Ma et al., 2024), or learning to learn, which optimizes model parameters across diverse learning tasks to enable rapid adaptation to new challenges; (ii) **Data-centric learning** (Li et al., 2020; Meng et al., 2023; Ma et al., 2024), which focuses on metric learning to compare distances between samples or expand synthetic data facing with data-scarce scenarios; (iii) **Transfer learning** (Tian et al., 2020; Luo et al., 2023; Zhang et al., 2024) where models pre-trained on large-scale datasets are fine-tuned on few-shot tasks to improve performance to improve performance by leveraging learned representations for more efficient adaptation.

The adaption-based methods (Sun et al., 2019; Ma et al., 2024) are densely connected to the model design, which attempts to directly establish a mapping function between input and prediction. By rapidly updating parameters on new tasks with a small number of samples, these methods facilitate the transfer of knowledge from previously learned tasks, making them highly effective in few-shot learning scenarios.

The data-centric methods utilize synthetic data (Meng et al., 2023) or metric learning (Ma et al., 2024) to adapt data-insufficient scenarios. The former involves generation methods like Generative Adversarial Networks (GAN) (Li et al., 2020) and auto-encoders (Yan et al., 2023b). These approaches generate additional training data to enhance the model's performance, mitigating the scarcity of labeled examples and improving generalization to new tasks. The latter, metric learning, tries to build data connections with the thoughts of nearby neighbors, focusing on learning a distance metric that clusters similar examples together and separates dissimilar ones. These data-centric methods emphasize the properties of the data to enhance the model's ability to generalize from a few examples.

The fine-tuning-based methods (Tian et al., 2020; Luo et al., 2023; Zhang et al., 2024) involve using pre-trained models on large datasets and fine-tuning them on the target task with a small number of examples. This approach leverages the knowledge gained from the pre-training phase and transfers it to the specific requirements of the new task. However, with the increasing amount of raw data on the Internet, it's challenging for pre-trained models, including Vision-Language Pretraining Models (VLM) (Zhu et al., 2024), to generalize to specific novel data, especially when the data is in different modalities.

Recent works on cross-domain learning (Wang et al., 2020; Li et al., 2022; Xu et al., 2023) also focus exclusively on learning from unimodal data. However, their limitations become apparent in complex real-world applications that often require understanding multiple modalities simultaneously. In this work, we extend its applicability of few-shot learning to real-world scenarios where data come from diverse visual modalities.

## A.2 RELATION TO MULTI-MODAL FEW-SHOT LEARNING

Existing multi-modal few-shot learning methods aim to leverage information from multiple modalities, such as combining visual data with textual (Xing et al., 2019; Tsimpoukelli et al., 2021; Alayrac et al., 2022; Lin et al., 2023c; Shao et al., 2024), audio (Meshry et al., 2021; Majumder et al., 2022; Kong et al., 2024), or tabular data Liu et al. (2022); Ye et al. (2024); Han et al. (2024). These approaches

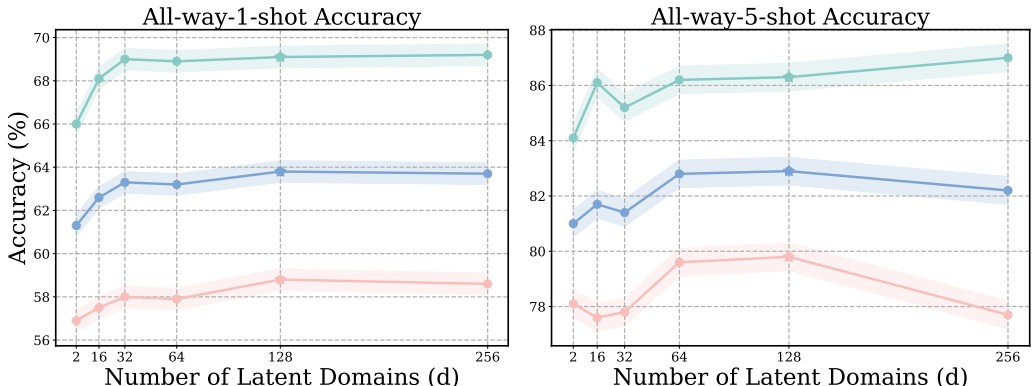

Figure 7: Hyperparameter analysis of the selection of the latent domain number $d$ on the SKETCHY dataset. The Blue line denotes the average accuracy of mixed modality data; the Green line stands for the average accuracy of RGB data; the RED line represents the average accuracy of sketch data.

focus on enriching the feature space by integrating heterogeneous data sources, thereby improving the generalization capabilities of few-shot models in scenarios where additional modality information is available. However, most of these methods concentrate on leveraging modalities that are inherently different from visual data, like text or audio, to provide semantic context or supplementary cues. They often assume access to richly annotated data in these auxiliary modalities, which may not always be feasible in practical applications. These methods enhance recognition by supplementing visual features with external information but do not address the challenges that arise when dealing with multiple visual modalities.

Few methods address the challenges of visual multi-modal scenarios where multiple visual modalities of the same object exist, such as RGB images, sketches, infrared images, or depth maps. Visual modalities often exhibit unique characteristics and structural properties, leading to significant domain gaps that complicate direct knowledge transfer between them. Some approaches Bhunia et al. (2022) utilize additional visual modalities to enhance recognition performance on a primary modality; however, they typically treat the additional visual modality as auxiliary information to support unimodal recognition rather than fully integrating multiple visual modalities into a unified learning framework.

In contrast, our work specifically targets the problem of Cross-modal Few-Shot Learning (CFSL) within the domain of visual data. We focus on recognizing instances across different visual modalities when only a few labeled examples are available. Our approach acknowledges that in real-world scenarios, models must often adapt to new visual modalities with limited annotated data, without the luxury of abundant multi-modal annotations. Unlike existing methods that leverage auxiliary modalities to aid a primary modality, we aim to develop a model capable of understanding and generalizing across diverse visual modalities in a few-shot setting.

In summary, while existing multi-modal FSL methods aim to enhance performance by integrating different data types, our work addresses the unique challenges of cross-modal recognition within visual data. We emphasize the importance of transferring knowledge from abundant unimodal data to novel visual modalities in a few-shot context, without relying on auxiliary modalities for support. Our approach better reflects real-world challenges and contributes a novel perspective to FSL.

## B   ADDITIONAL EXPERIMENTS

We conducted a hyperparameter analysis to examine the impact of the number of latent domains $d$ on our model's performance using the Sketchy dataset under the All-way 1-shot and 5-shot settings. Figure 7 illustrates how varying $d$ influences the accuracy across different modalities: mixed modality data, RGB data, and sketch data, with results averaged over multiple trials.

For **mixed modality data** (blue line), the accuracy remains relatively stable as $d$ increases, with slight improvements observed up to $d = 128$. Beyond this point, performance plateaus, indicating

Table 4: The details of the proposed GTL framework architectures. BS is short for batchsize, BN is short for BatchNorm1d. d determines the number of latent domains.

| Module | Description | Dimenssions |
|---|---|---|
| Encoder Input: visual representation $\mathbf{x}$ | | BS $\times$ 1280 |
| Dense | 256 neurons, with BN, ReLU, Dropout | BS $\times$ 256 |
| Dense ($\mu$) | mean of posterior ($N_c + N_m$) neurons | BS $\times (N_c + N_m)$ |
| Dense ($\sigma$) | variance of posterior ($N_c + N_m$) neurons | BS $\times (N_c + N_m)$ |
| Reparameterization | Sampling | $\hat{\mathbf{z}}_c$ ($N_c$) + $\hat{\mathbf{z}}_m$ ($N_m$) |
| Disturbance encoder Input: visual representation $\mathbf{x}$ | | BS $\times$ 256 |
| Gate | Learnable gating function | BS $\times$ d |
| Dense | d * $N_m$ neurons | BS $\times$ d $\times N_m$ |
| Combination | Element-wise weighted sum | BS $\times N_m$ |
| Additional Aggregator Input: latent $\hat{\mathbf{z}}_m$ | | BS $\times N_m$ |
| Dense | Aggregation, $N_m$ neurons | $\hat{\mathbf{z}}'_m (N_m)$ |
| Decoder Input: Concat ($\hat{\mathbf{z}}_c, \hat{\mathbf{z}}'_m$) | | BS $\times (N_c + N_m)$ |
| Dense | 256 neurons, with BN, ReLU, Dropout | BS $\times$ 256 |
| Dense | 1280 neurons | BS $\times$ 1280 |
| Classifier Input: latent intrinsic concept $\hat{\mathbf{z}}_c$ | | BS $\times N_c$ |
| Dense | 1280 neurons | BS $\times$ 1280 |
| Dense | Classification output | BS $\times$ Class Number |

diminishing returns from increasing the number of latent domains further. The **RGB data** (green line) shows consistently high accuracy in the 1-shot setting, with minor improvements up to $d = 64$ in the 5-shot setting, after which performance stabilizes. The **sketch data** (red line) exhibits more variability, especially in the 5-shot setting, where accuracy decreases at intermediate values of $d$ but recovers and improves as $d$ approaches 128. This suggests that the model's ability to capture sketch-specific features is sensitive to the number of latent domains, particularly when fewer domains are considered.

We selected $d = 128$ as the optimal number of latent domains for several reasons. First, at $d = 128$, the performance across all modalities is near its peak, which is crucial for our CFSL task, which relies on effectively handling multiple modalities. Second, the larger domain size helps stabilize the variability observed in sketch data, especially in the 5-shot setting, leading to more robust performance across both 1-shot and 5-shot scenarios. Lastly, while larger values of $d$ (e.g., $d = 256$) do not offer significant performance gains, they introduce additional computational overhead without clear benefits. Thus, $d = 128$ strikes a balance between performance and efficiency, enabling the model to effectively capture latent shared concepts across modalities.

Notably, the effect of increasing $d$ is less pronounced in low $k$-shot settings, particularly for smaller values of $k$. This diminished effect may be due to the limited number of latent domains being insufficient to capture the complex variations in the data when only a few examples per class are available. With small $k$, the model has less data to inform the latent space, making it challenging to effectively utilize a larger number of latent domains. Conversely, too many latent domains relative to the limited data can lead to overfitting or poor generalization. This highlights the importance of carefully selecting the number of latent domains in relation to the available data and the specific characteristics of each modality to optimize performance in few-shot learning scenarios.

## C  NETWORK ARCHITECTURES

Table 4 provides a comprehensive overview of our GTL framework's architecture. We empirically set the intrinsic concept dimensionality $N_c$ as 128 and the modality-specific disturbance dimensionality $N_m$ as 64 based on preliminary experiments that balanced model expressiveness and computational efficiency. The number of latent domains $d$ is set to 128, aligning with our hyperparameter analysis, indicating optimal performance at this value.

**Random Seed.** To ensure the reproducibility of our experiments, we set the random seed to 0 for all runs.

