# OpenReview forum: "Cross-Modal Few-Shot Learning: a Generative Transfer Learning Framework"
_ICLR.cc/2025/Conference — ICLR 2025 Conference Withdrawn Submission_

### Official Review · Reviewer_2TGd · 2024-11-03

**Soundness:** 3
**Presentation:** 3
**Contribution:** 3
**Rating:** 6
**Confidence:** 4

**Summary:**

In this paper, the authors introduce a new few-shot task, Cross-modal Few-Shot Learning (CFSL), which aims to recognize instances from multiple modalities when only a few available labeled examples. Accordingly, this paper proposes a Generative Transfer Learning (GTL) framework for this new task. GTL is designed to disentangle intrinsic concepts from in-modality disturbances, enabling efficient knowledge adaption across modalities. Experiments on four datasets validate the effectiveness of the proposed method.

**Strengths:**

1.	The paper is overall well written and easy to follow, making it comfortable to read. The authors do not engage in unnecessary technicalities and provide the required background in a succint and clear manner.

2.	The proposed CFSL task is novel (to the best of my knowledge) and a valuable contribution to the vision community.

3.	The proposed GTL framework is novel and effective in solving the CFSL problem regarding RGB images and sketches.

**Weaknesses:**

1.	The experiments conducted by the authors on CFSL tasks only involve RGB images and sketches, which is not sufficient to demonstrate GTL's generalization in other multimodal recognition tasks, such as infrared images.

2.	This paper adopts a generative framework, but there is no presentation or analysis of the generated results. More visualization and analysis of the generated results are desirable.

3.	It seems that the proposed method can also be used for unimodal few-shot learning. It would be interesting to see how it performs on this task.

**Questions:**

One important assumption of this paper is that the relationship between latent representations and visual content remains consistent across both base and novel datasets, hence the parameters of the generator in Phase 2 is fixed. The author demonstrated this through experiments, but I think it might only hold true when multimodal representations are obtained by the same encoder. In fact, multimodal data varies significantly and often requires different encoders, so whether the framework proposed in the paper is effective under these conditions still needs to be verified.

---

### Official Review · Reviewer_D2Sf · 2024-11-03

**Soundness:** 2
**Presentation:** 2
**Contribution:** 1
**Rating:** 3
**Confidence:** 4

**Summary:**

This paper introduces Cross-modal Few-Shot Learning (CFSL), aiming to recognize instances from multiple visual modalities with limited labeled examples. The authors propose a Generative Transfer Learning (GTL) framework to disentangle shared concepts across modalities from modality-specific variations. The framework uses a two-stage training process: learning latent concepts from unimodal data, then adapting to novel multi-modal data. Experiments on cross-modal datasets show improvements over existing methods.

**Strengths:**

The paper addresses an important challenge in few-shot learning across visual modalities.

The proposed GTL framework offers an interesting approach to disentangling shared concepts from modality-specific variations.

The experimental results show some improvements over baseline methods on the chosen datasets.

**Weaknesses:**

Limited scope of multi-modality: Despite claiming to address multi-modal learning, the paper focuses primarily on RGB images and sketches. This narrow focus doesn't fully align with the broader multi-modal challenges described in the introduction, such as video or other visual modalities.

Lack of comparison with state-of-the-art few-shot methods: The paper doesn't provide sufficient evidence that existing few-shot learning methods fail in cross-modal scenarios. There's no comparison with recent advanced few-shot learning techniques, such as "Context-Aware Meta-Learning," which has shown promise in cross-domain few-shot learning.

Insufficient justification of the task setting: The paper doesn't adequately differentiate the proposed CFSL task from existing cross-domain few-shot learning problems. It's unclear whether this truly represents a novel challenge or is simply a reframing of known issues.

Limited theoretical foundation: The paper lacks a strong theoretical basis for why the proposed method should work better than existing approaches in cross-modal scenarios.

Narrow experimental evaluation: The experiments are limited to a small set of visual modalities and don't explore the full range of multi-modal challenges suggested in the introduction.

**Questions:**

How does the proposed CFSL task fundamentally differ from existing cross-domain few-shot learning problems?

Can you provide empirical evidence showing that state-of-the-art few-shot learning methods (e.g., "Context-Aware Meta-Learning") fail in the proposed cross-modal scenarios?

Why does the experimental evaluation focus only on RGB images and sketches when the introduction suggests a broader range of visual modalities?

Can you provide a theoretical analysis or justification for why the GTL framework should outperform existing methods in cross-modal few-shot learning?

How would the proposed method perform on more diverse multi-modal datasets that include other visual modalities like video or depth maps?

---

### Official Review · Reviewer_ZKbS · 2024-11-04

**Soundness:** 3
**Presentation:** 3
**Contribution:** 3
**Rating:** 6
**Confidence:** 4

**Summary:**

This paper introduces a new cross-modal few-shot learning (CFSL) task, aimed at classifying instances with multiple visual modalities using a limited number of labeled samples. Unlike traditional supervised learning and single-modality few-shot learning, CFSL combines data from multiple visual modalities, adding complexity due to inherent variability and domain differences among them. To address these challenges, the authors propose a generative transfer learning (GTL) framework, which decomposes the target object's latent concepts into cross-modal shared intrinsic concepts and modality-specific perturbations. The GTL framework involves two training stages: the first focuses on capturing intrinsic concepts and shared variations across modalities, while the second trains separate classifiers to predict labels based on the learned latent intrinsic concepts. Extensive experiments on multiple cross-modal datasets (including SKETCHY, TU-BERLIN, MASK1K, and SKSF-A) validate the effectiveness of the proposed approach.

**Strengths:**

- The introduction of the CFSL task presents a fresh research direction that goes beyond traditional single-modal few-shot and supervised learning. The GTL framework innovatively addresses cross-modal variation and domain discrepancy by decomposing the latent concepts of target objects into cross-modal intrinsic concepts and modality-specific perturbations.
- Extensive experiments conducted on various cross-modal datasets provide a comprehensive comparison with state-of-the-art methods, fully verifying the proposed method's effectiveness. The authors also offer detailed discussions and validation of their assumptions through visualization analysis.
- The paper is well-organized and clearly written, with comprehensive descriptions of the problem definition, methodology, and experimental design. The inclusion of illustrative figures and data analyses further aids reader comprehension of the core concepts.

**Weaknesses:**

- The paper notes that existing multimodal datasets focus predominantly on limited visual modalities, such as RGB images and sketches. This limited scope restricts the model's potential in more complex visual contexts. Future work should consider broader, more diverse multimodal datasets to fully explore and validate the potential of cross-modal few-shot learning.
- Figure 5 illustrates the motivation behind the proposed method, specifically the learning of distinct concepts. However, additional intuitive examples, such as comparisons between local image patches and learned concepts, would help readers gain a clearer understanding of the specific content of concept learning.
- Some relevant discussions related to concepts and few-shot learning are overlooked, including recent works like:

[1] Concept Learners for Few-Shot Learning. ICLR 2021.

**Questions:**

Please see the Weaknesses

---

### Official Review · Reviewer_Z6Rv · 2024-11-04

**Soundness:** 2
**Presentation:** 2
**Contribution:** 2
**Rating:** 5
**Confidence:** 3

**Summary:**

In this paper, the authors introduced the Cross-modal Few-Shot Learning (CFSL) benchmark that aims to recognize instances from multiple modalities in a data efficiency setup. To tackle this challenge, they proposed the GTL, a two-step approach that initially involves training on extensive unimodal datasets, followed by a transfer learning phase to effectively adapt to novel concepts. The key idea is to learn shared latent representation across multiple modalities while modeling variations inherent within each modality. Experimental results from four multimodal datasets demonstrate that this approach outperforms state-of-the-art methods, enabling generalization to unseen modalities with only a few samples.

**Strengths:**

+ Proposing the interesting idea of transfering knowledge from large-scale unimodal data to data-scarce multi-modal scenarios for an important task of cross-modal few-Shot learning
+ The paper is clear and well written, but the insights on the effectiveness of the proposed method should be better discussed, explained, and justified
+ Experiments are done on multiple benchmarks, but the comparison could be more complete on more complex multimodal datasets

**Weaknesses:**

- Contributions are not clearly and accurately stated, and there is a lack of enough methodological originality (e.g., the method seems complex combinations of multiple loss component with multi-phase training, etc.)
- There is a lack of motivation and discussion on the proposed solution
- There is a clear lack of in-depth theoretical analysis of the proposed method
- Poor English

**Questions:**

Although the proposed method is potentially interesting, the paper fails to clearly show the benefit of the proposed method in few-shot setup. I believe even complex methods in such interesting new settings can and should be published even if they do not outperform the state-of-the-art methods. It’s a duty of authors, however, to thoroughly demonstrate the pros and cons of such methods and the impact of each component, so that the reader can learn from them.

Details:

- More in-depth discussion of the method is necessary, For example: Why does it work? When does it fail?
- Theoretical discussion is missing: there is no theoretically evidence provided to support why each loss componnet is nesseary
- Providing experimental results on more complex multimodal datasets would be very helpful
- English should be improved

---

### Note · Authors · 2024-11-15

**Comment:**

Thank you to all the reviewers for your insightful and constructive feedback on our paper. We have carefully considered your comments and have decided to withdraw our submission at this time to refine our work further. Your efforts have been greatly appreciated and will guide our improvements.

**Withdrawal Confirmation:**

I have read and agree with the venue's withdrawal policy on behalf of myself and my co-authors.